# Phase-In to Phase-Out—Targeted, Inclusive Strategies Are Needed to Enable Full Replacement of Animal Use in the European Union

**DOI:** 10.3390/ani12070863

**Published:** 2022-03-29

**Authors:** Lindsay J. Marshall, Helder Constantino, Troy Seidle

**Affiliations:** Humane Society International, Avenue des Arts 50, 1000 Bruxelles, Belgium; hconstantino@hsi.org (H.C.); tseidle@hsi.org (T.S.)

**Keywords:** new approach methodologies, animal replacement, biomedical research

## Abstract

**Simple Summary:**

In the European Union (and elsewhere), the overall use of animals in laboratories has failed to undergo any significant decline, despite six decades of purported adherence to the “3Rs” principles of replacement, reduction, and refinement. In the EU, the 1986 adoption of a legal requirement to use scientific methods not entailing the use of live animals, rising public opinion against the use of animals and the almost exponential rise in development and application of non-animal new approach methodologies (NAMs) signals a readiness to end animal testing. Indeed, the European Parliament recently carried an almost unanimous vote to adopt an action plan to phase out the use of animals in research and testing. This article explores what is needed to make this action plan a success, considering all stakeholders and their needs.

**Abstract:**

In September 2021, the European Parliament voted overwhelmingly in favour of a resolution to phase out animal use for research, testing, and education, through the adoption of an action plan. Here we explore the opportunity that the action plan could offer in developing a more holistic outlook for fundamental and biomedical research, which accounts for around 70% of all animal use for scientific purposes in the EU. We specifically focus on biomedical research to consider how mapping scientific advances to patient needs, taking into account the ambitious health policies of the EU, would facilitate the development of non-animal strategies to deliver safe and effective medicines, for example. We consider what is needed to help accelerate the move away from animal use, taking account of all stakeholders and setting ambitious but realistic targets for the total replacement of animals. Importantly, we envisage this as a ‘phase-in’ approach, encouraging the use of human-relevant NAMs, enabling their development and application across research (with applications for toxicology testing). We make recommendations for three pillars of activity, inspired by similar efforts for making the shift to renewable energy and reducing carbon emissions, and point out where investment—both financial and personnel—may be needed.

## 1. Introduction

In September of 2021, the European Parliament voted overwhelmingly in favour of a resolution to phase out animal use for research, testing and education, through the adoption of an action plan [1]. This resolution stems from several observations. In the European Union (and elsewhere), the overall use of animals in laboratories has failed to undergo any significant decline (Figure 1), despite enshrining the “3Rs” principles of replacement, reduction and refinement in the European Directive 2010/63/EU on the Protection of Animals for Scientific Purposes; (Article Four ‘Principle of replacement, reduction and refinement’ [2]); rising public opinion against the use of animals with almost three-quarters of Europeans in favour of phasing out animal testing [3] and the increase in development and application of non-animal new approach methodologies (NAMs) [4,5,6]. The resolution acknowledged the role that animal research has played, stating “whereas previous animal testing has contributed to advances in developing treatments for human health conditions, as well as medical devices, anaesthetics and safe vaccines, including COVID-19 vaccines, and has also played a role in animal health” but also, importantly, reflects the rise in development and possibilities of the non-animal approaches: “whereas the toolbox of non-animal testing models is growing and shows the potential to enhance our understanding of diseases and accelerate the discovery of effective treatments; whereas this toolbox includes, for example, new organ-on-chip technology, sophisticated computer simulations, 3D cultures of human cells for drug testing and other modern models and technologies”. As the Parliament’s resolution text further acknowledges, this means that the vision in the European Directive 2010/63/EU on the Protection of Animals for Scientific Purposes to reach a “full replacement of procedures on live animals … as soon as it is scientifically possible to do so” could be accelerated in the EU. Thus, despite strong language advocating for the replacement of animals, it is clear that existing legislation and strategies are not sufficient to achieve the shift.

Here we explore the opportunity that the action plan could offer in developing a more holistic outlook for fundamental and biomedical research, which accounts for around 70% of all animal use for scientific purposes in the EU [7]. We specifically focus on biomedical research, which we define here as research that is **not** carried out to satisfy any regulatory requirement(s). We consider how mapping scientific advances to patient needs (considering the vital role of basic science in feeding the discovery pipeline [8] and the possible advantages to the inclusion of patients in basic research [9]), taking into account the ambitious health policies of the EU, would facilitate the development of non-animal strategies to deliver safe and effective medicines, for example. We consider what is needed to help accelerate the move away from animal use, taking account of all stakeholders including scientists, irrespective of their animal use, the public and patients, and setting ambitious but realistic targets for total replacement of animals. Importantly, we envisage this as a ‘phase-in’ approach, encouraging the use of human-relevant NAMs, enabling their development and application across research (with applications for toxicology testing). We make recommendations for three pillars of activity, inspired by similar efforts for making the shift to renewable energy and reducing carbon emissions, and point out where investment—both financial and personnel—may be needed.

## 2. The Current Situation—Sustained Reliance on Animals across the Research Spectra

European Directive 2010/63 on the Protection of Animals for Scientific Purposes articulates the need for animal welfare at its heart and throughout, stating “the final goal of full replacement of procedures on live animals for scientific and educational purposes as soon as it is scientifically possible to do so…”. However, it is apparent when studying the available data on animal use for scientific purposes (Figure 1 and see also ALURES database) that effective implementation of the Directive alone is not sufficient to drive meaningful reduction, let alone full replacement.

For the purpose of this commentary, we focused on the use of animals for biomedical research. For this, we combined data on animal use for “Basic research” with animal use for “Translational and applied research”, as recorded in the ALURES database. ALURES was launched in 2020 and collates annual data on animal use across the EU, starting with data from 2015. In contrast to animal use for “Regulatory use and Routine production”, basic research and translational and applied research are not associated with a legal mandate to use animals and are therefore may be likely to capture “experimental” animal use, prior to any regulatory submission. Although the Directive does not provide a definition of these categories, statistical returns indicate that animals used for oncology, study of organ systems, sensory organs, metabolism, multisystemic, and animal behaviour are returned under the category ‘Basic research’. For translational and applied research, animal use includes human diseases, animal diseases, animal welfare, disease diagnosis and non-regulatory toxicology and ecotoxicology. However, in combination, animal use for basic research and applied and translational research accounts for almost 70% of the total animal use, representing around seven million animal uses each year, and has done for the past four years for which such records are available online, through the ALURES database (2015–2018). We focused on the data in ALURES as this is freely available although this has limited our analysis to animal use between 2015 and 2018. However, we note that some member states have published additional statistics on animal use for 2019, but these reports are not available for all member states that contribute to ALURES and therefore we used ALURES as the most complete dataset. Note that earlier data on animal use are available through the European Commission portal at https://ec.europa.eu/environment/chemicals/lab_animals/reports_en.htm (accessed on 20 February 2021).

Figure 1 summarises the data on animal use for Basic research, Translational and applied research, and Regulatory use, submitted to the European Commission and available through the ALURES database. These data illustrate the fact that reliance on animals as surrogates for humans in biomedical research (represented by Basic research plus Translational and applied research; grey bars) has not significantly declined over this timeframe. The reasons underpinning this lack of a sustained decrease are likely complex, possibly controversial and multi-factorial, and a detailed analysis of them is beyond the scope of this commentary. Briefly, animal use may be justified in terms of the possible benefit to human health, animal health, or the environment, but the evaluation of potential benefits is problematic, with a tendency to “over-promise” the likely advantage (to humans) [10]. Additionally, the choice of animal model has been shown to depend on the historical use of animals rather than the most valid scientific approach [11]. Perhaps it is not surprising that a survey of animal researchers revealed their perception of several roadblocks hampering the development and implementation of animal-free tools [12], and this may be (partly) associated with a lack of formal education in, and exposure to, the innovative, non-animal approaches, as we discuss later. Thus, we suggest that all of these factors, and probably more, are underpinning the continued use of animals.

Reuse of animals has accounted for around 2% of total animal use for each year in which reuse data are available (2015 to 2018, via ALURES database), and therefore does not significantly impact the observed lack of decline. It is also not appropriate to suggest that increasing animal re-use be employed as a strategy in order to reduce the absolute number of animals. Animal reuse is rightly, strictly governed, and must occur on a case-by-case basis. The Directive states that “the benefit of reusing animals should be balanced against any adverse effects on their welfare, taking into account the lifetime experience of the individual animal” [2].

The data presented in blue bars in Figure 1 show the total animal use for all purposes—namely research, testing and education. The data in Figure 1 also suggest that animal use for regulatory use and routine production (grey bars) has been stable at around two million uses per year, and this is true for the last decade (data not shown). Note that there are differences in data submission (reporting requirements changing, misunderstanding of requirements, the addition of Member States) that make accurate annual comparisons challenging, but even with this variability, there is no indication that animal use is undergoing a sustained significant decrease, despite the clear legal requirement laid out in the 2010 Directive and its predecessor. Note that the increase in animal use from 2017 to 2018 is in part due to the contribution of data from Norway for the first time.

The animals most commonly used for research and testing are rodents and fish. However, mice bear the brunt of biomedical research (Figure 2), for reasons of scientific tractability (e.g., ease of genetic manipulation) and cost-effectiveness, rather than scientific rationale. In fact, for translational and applied research at least, the availability of the (animal) model is more likely to underpin model choice than similarity to human pathology [11]. On average, between 2015 and 2018, over 4.5 million uses of mice were recorded each year for biomedical research (defined here as animal use for Basic research combined with animal use for Translational and applied research according to ALURES categories). This is in contrast to mouse use for Regulatory use and Routine production (dotted line, Figure 2), which has undergone a year-on-year decrease and represents less than 10% of all animal use—whereas mouse use for basic, applied and translational research remains at just under 50% of all animal use.

The most recent data collected for the 28 Member States of the European Union and Norway are from 2018 (when the United Kingdom was still part of the European Union and therefore UK data are included in these statistics). These data indicate that 4.46 million mice were used for biomedical research, with a further one million uses of mice for regulatory use and routine production. When calculated as a percent of all animal use for all purposes, we see that biomedical research using mice, at around 80% of total mouse use, represents the vast majority with no signs of significant decline since the introduction of the Directive.

The decrease of almost half a million mice between 2017 and 2018 (when taking account of the 28 Member States only and not considering the data from Norway), appears promising. However, scrutiny of the changes in animal use indicates fluctuations of hundreds of thousands in either direction across the years—further evidence of the lack of a sustained downward trend. Between 2015 and 2017 there were no changes made to reporting requirements or number of Member States submitting data, and the number of animals did not change significantly in either direction during this time period; thus, it would be a major assumption that a decrease in animal use for one year (as seen between 2017 and 2018) will translate to a sustained, cumulative decrease over many more years without accompanying policy efforts. One of the reasons for this could be that, despite a vision of full replacement of procedures on live animals for scientific purposes, Directive 2010/63/EU focuses its attention on raising animal welfare standards and providing rules governing the use of animals in scientific procedures. This includes, for instance, stating provisions on the use of certain animals, such as non-human primates and endangered species; guidelines on classification of the severity of procedures; rules on anaesthesia; and project evaluation and authorisation.

Directive 2010/63/EU is required to encourage consideration of raised standards for animal welfare across the EU, but it appears that implementation of the Directive alone is not enough to address the scientific challenge of entirely moving away from animal use. Article 47 “Alternative approaches” is the only article in the sixty-six that comprise the Directive that specifically describes the requirement for application of the 3Rs. It is therefore timely that the Directive is complemented with an ambitious and proactive action plan. The Parliament’s near-unanimous Resolution from September 2021 offers the opportunity to do just that, and to achieve a sustained decrease in animal use while contributing to the EU health initiatives by accelerating the shift to more human-relevant approaches to research into health and disease.

## 3. Transitioning Biomedical Research towards Human-Based, Non-Animal Methods Represents an Essential Step in Achieving the EU’s Public Health Objectives

The EU health research initiatives https://ec.europa.eu/info/research-and-innovation/research-area/health-research-and-innovation_en (accessed on 4 January 2022) encompass several key research areas where the human-relevant approaches are already proving illuminating. For example, for disparate conditions ranging from cancer [13,14,15] through brain diseases [16,17] to rare diseases [18], researchers are successfully applying methodologies no longer reliant on the use of animals as ‘disease models’ in order to develop potential new treatment options and offer a clearer understanding of the human condition.

Rare diseases offer an ideal opportunity to harness the impressive advances in genetic and technological science over the last decades for the identification of new “druggable” targets, to understand the mechanism(s) of disease progression and for faster repurposing of existing drugs as potential treatment options for people living with these conditions. The ongoing COVID-19 pandemic has revealed the potential for microphysiological systems (organ chips) to provide fast and effective drug repurposing [19]. It may be possible to apply organ chips, developed using patient cells, to screen approved drugs and discover potential personalised treatments. As research using the chips, and advances in other cell-based NAMs, evolve so that knowledge and experience accumulate, it becomes apparent that chips can be used to measure increasingly complex parameters such as immune cell recruitment [20,21]; thus, these modern, human biology-based NAMs offer a useful platform to look at drug/combination efficacy as well as safety.

It is also apparent that these rare diseases do not lend themselves to the creation and use of genetically altered animals as surrogates for those diseases genetic in origin, which comprise the vast majority of rare diseases [22]. This necessitates a different approach, whereby data mining [23], existing data [24], and use of patient biological materials [25,26] are utilised for “pre-clinical” testing, instead of expensive and time-consuming animal models. As one recent success illustrating where NAMs may help to decipher rare diseases, Chou and colleagues developed a human bone marrow chip that recapitulated hematopoiesis [27]. Importantly, chips created using cells from patients with the rare disease Schwachman-Diamond syndrome demonstrated a hypoplastic phenotype, with impaired maturation and aberrant surface marker expression of neutrophils—mimicking the neutropenia and other clinical aspects of the disease. Screening drugs in the chips could be a time- and cost-effective way to get much-needed treatments to the people who require them, and could offer a route to personalised medicine.

Overall, the European Commission, through its European Health Initiatives and funding programmes, recognises the inability of a single country or methodology to address these issues. In the area of brain research, for example, there are several initiatives designed collectively to address the need for a better understanding of brain function and also to diagnose and treat brain diseases. However, threaded throughout these projects there remains a reliance on in vivo models employing rodents, and thus a failure to fully embrace human-centric methodologies. This despite mounting scientific evidence that rodent brains fail to fully recapitulate the complexity or functionality of the human organ [28,29], and even declarations that “… species-specific features emphasize the importance of directly studying the human brain.” [30].

Nonetheless, there is much to celebrate in these programmes. One example is captured in the strategy for the EU Joint Programme on Neurodegenerative Disease Research (JPND), which is the largest global research initiative aimed specifically at neurodegenerative diseases. The JPND Research and Innovation strategy includes several workplans focused on the use of patients, patient data, phenotypic screening and patient stratification, along with the development of cellular methods using human tissue or stem cells, [31] indicating the value of these non-animal approaches in addressing the issues. However, there remains some reliance on “improving” animal models and an emphasis on “reverse translation”, despite countless failures and the dismal translational success of animal models of neurodegenerative disease. In 2019 alone, 132 agents were in a clinical trial for the treatment of Alzheimer’s disease [32], yet the treatment options remain extremely limited, with either high costs [33] or adverse side effects [34], proving disadvantageous to patients and carers.

Currently, preclinical testing, the stages of drug development that occur prior to testing in human volunteers or patients, relies heavily on animal models, and it is no coincidence that—partly due to the insurmountable species differences between rodents, dogs and humans—current drug failure rates are around 95%, often as a result of unexplained toxicity or lack of efficacy [35,36]. It is therefore likely that more innovative human biology-based approaches can offer a more relevant, predictive and cost-effective path for preclinical drug discovery. The pharmaceutical industries should be applauded for their efforts in supporting the development and use of NAMs [37], but it seems that their hands are effectively tied against the greater use of these methods until regulations prioritise NAMs above animal data. As stated in the JPND research and innovation strategy: “To accelerate translation of basic findings to clinical benefit, the validity of model systems used for target identification and therapeutic development needs improvement” [38]. This should be accompanied by a clear acknowledgement as to when and where specific animal models are failing to translate, and a commitment to no longer fund further research using these models. The former Innovative Medicines Initiative (IMI) noted the need to “eliminate poorly predictive animal models”, including those for Parkinson’s, depression, autism and schizophrenia [39], yet continued to fund projects developing or applying animal models in these areas. A review of the closed calls for IMI2 (https://www.imi.europa.eu/apply-funding/closed-calls (accessed on 7 January 2022)) reveals many methods and tools that employ human cells, patient data, in silico tools and thus an increase in research projects and programmes using non-animal models to improve the predictivity and human relevance of these important topics. However, it is also true that many of these non-animal models are being developed and used in parallel to the animal models, despite the acknowledged limitations of the animal-based approaches and their inability to translate.

Despite some efforts to invest in more human-relevant approaches, particularly within the portfolio of work funded by IMI and IMI2, there remains a need to define human-based, non-animal models as the preferred method of investigation in health research wherever possible. In June 2021, the EU Innovative Health Initiative was announced [40] as the successor of the IMI for the advancement of medical technology, digital health and diagnostics. Among its strategic priorities is to “… strive to pursue the aims of Directive 2010/63/EU on the protection of animals used for scientific purposes and, in particular, the principle of the Three Rs to replace, reduce and refine the use of animals.” [41]. However, there remains a lack of clear emphasis on prioritised funding for human biology-based non-animal approaches throughout the research agenda. The adoption of unambiguous language prioritising development and application of human-based, non-animal tools in all funding calls will help to accelerate further development of human-predictive technologies, appreciation of where more input may be required (i.e., gap analysis), and further increase confidence and familiarity with the processes and data that will help to embed these non-animal approaches—and by extension animal replacement—in biomedical research across the EU.

## 4. Planning an Inclusive Transition to Non-Animal Research

A transition away from tradition towards innovation is nothing new— it is what humans have been doing since they began to walk upright—and therefore shifting from the use of animals as models in research and testing to more human-relevant, non-animal approaches can be seen as one more advance. Although historically, research on animals has offered a repository of scientific information which has been applied to better our understanding of human (patho)physiology, there are still many unknowns that require human biology-based approaches. Additionally, of course, animals have been used for the development of effective medicines, since all drugs have to be tested on animals, but it seems timely to consider what we might be missing—where drugs that are toxic to animals are lost from the development pathway [42]—and so we are calling for a more rapid shift towards animal replacement. Also, we can draw from previous experiences and ongoing initiatives to map out a safe, effective and inclusive route toward full replacement that ensures that all stakeholder needs and requirements are considered (Table 1). The concept of a “Just Transition” was recently articulated by the European Commission with regard to its intention to make the move away from fossil fuels in the context of the European Green Deal [43]. This was followed by an announcement from the European Research Executive Agency of a budget increase to 111 million EUR with which to support collaborative research and the development of breakthrough technologies to enable the shift away from fossil fuels [44].

Of course, the shift away from using animals in research and testing is not on the same scale as the move to net-zero emissions. However, even with this change in magnitude, there are similarities in terms of the intentional, holistic and methodical approaches applied for carbon neutrality that could be adopted to initially reduce, and ultimately replace, animal use. In line with other initiatives from the European Commission, we are suggesting that targeted reduction in animal use could be divided into three pillars: (Pillar 1) “Promoting innovative science with human biology as the gold standard” encompasses the scientific and technological advances underpinning the development and use of advanced, human-based non-animal NAMs; (Pillar 2) “Agile regulations” moves beyond fundamental biomedical science to address what is required to drive increasing confidence, trust and use of NAMs; (Pillar 3) “Knowledge transfer” includes the vital elements of education and (re)training that are needed to create a fit-for-purpose workforce and to enable stakeholders invested in animal-based research to pivot away from this without losing their livelihoods.

To facilitate an inclusive transition away from animal models in biomedical research and toward the more human-relevant tools, it is important to consider all the stakeholders, beyond the researchers themselves. There is also the need to consider the wider population, who may be destined to become patients and therefore to think of disease prevention strategies, which could have an indirect impact on animal use but are more health policy-focused, so are not included here as part of our science-led Pillars. However, we recognise that many of the IHI future plans are placing patients at the centre of their strategies and that this, together with ensuring adequate support for geographical areas with either higher disease burden or reduced public spending capacity, are vital for improving lifestyle education and disease prevention.

Thus, Table 1 is not an exhaustive list, for example, besides patients with life-limiting or life-threatening conditions, we must also consider the stakeholder with most to lose in this—the animals themselves—and thus the ultimate purpose of adoption of these pillars is to ensure multi-stakeholder co-operation to enable full replacement of animals across the research and testing spectra. Table 1 compiles many of these actors, suggests the pillars into which they would fit and offers some suggestions of what may be needed to encourage the transition.

## 5. Pillar 1—Promoting Innovative Science with Human Biology As the Gold Standard

Meaningful progress toward the replacement of animals in the EU is unlikely to occur until the European Commission and Member States formally recognise that human biology-based tools and methods are the quintessential model for human health research. This recognition should then be reflected in all strategic science priorities, funding calls, and grants awarded. This would lead to a natural redirection of funding away from animal models with low human predictivity/translation towards more predictive, human-relevant NAMs.

Within the European Union’s most recent funding programme, Horizon Europe (https://ec.europa.eu/info/research-and-innovation/funding/funding-opportunities/funding-programmes-and-open-calls/horizon-europe_en (accessed on 15 December 2021)), 270 million euros funding was directed toward NAMs [45,46]. This equates to roughly 45 million euros per year across the six-year programme. Assuming a modest annual redirect of 5% earmarked specifically for human-based NAMs methods and research infrastructures would create a cumulative shift of nearly 120 million euros by 2040. A more ambitious redirect of 10% per annum would create a NAMs budget of 187 million euros by 2035. Both of these options are still modest (and therefore easily achieved) given overall Horizon Europe funds of €95.5 billion [45], and taking into account that all framework programmes have increased their overall budgets since their creation in 1984. It appears that investment in NAMs lags behind somewhat: while the overall budget from Framework Programme 7 to Horizon 2020 (which were the funding programmes prior to Horizon Europe and were active between 2007–2013 and 2014–2020, respectively) increased by 40% [47], spending on NAMs has remained stable over the same period according to the Commission [48]. With the budget from Horizon 2020 to Horizon Europe increasing again by almost 24%, it is time for more substantial investment in NAMs [49].

More clarity is needed to fully and fairly assess the EU’s budgetary commitments to NAMs. Within the CORDIS database, which collates EU research funding, there is currently no tracking mechanism to identify grants awarded to animal compared to non-animal research. A sound funding strategy will require tracking mechanisms to be developed and implemented. In addition, an approach based on the one described by the IMI which will “eliminate poorly predictive animal models” should be formally adopted in order to promote the development and use of human biology-based tools, promoting translation and maximising return on investment.

Along with restricted, ring-fenced funding for NAMs, a shift in application “focus” may be necessary here to ensure success. This could mirror recent calls for multi-disciplinary research applications that have seen engineers, mathematicians, clinicians, and biologists, etc., collaborating to great success [50,51,52]. Offering specific funding for collaborations between the existing NAMs developers and users with those researchers who need to make the transition could create the appropriate incentives, and confidence, to drive the shift away from animals. In addition, creating pools of experts on NAMs approaches who can critique these applications will be vital in ensuring that European science remains cutting edge and will provide the necessary return on investment.

There may also be a need for change in infrastructure and funding programmes should reflect this. As the transition away from animals occurs, there will be a greatly reduced need for dedicated animal facilities. Initially, identifying opportunities for resource sharing (under Pillar 3) in line with recent advances in the UK [53] and according to other initiatives such as ShARM [54] and SEARCHBreast [55,56] could begin the shift to reduced animal use without compromising research or careers during the initial stages of the transition. Ultimately, however, it would be advantageous to provide infrastructure grants to allow full conversion of the animal facilities, perhaps to create Centres of Excellence for human-relevant research. For example, in 2019, the Centre for Predictive Human Model Systems (CPHMS; https://aic.ccmb.res.in/cphms/ (accessed on 12 February 2022)) was set up by the Government of India’s Atal Incubation Centre—Centre for Cell and Molecular Biology (AIC-CCMB) in collaboration with Humane Society International/India, as India’s first think tank for human-relevant methods.

Success for the activities under Pillar 1—driving widespread acceptance of human biology as a gold standard—will require coordinated incentives for researchers to transition away from animals, through dedicated financial support for NAMs alongside education and training.

## 6. Pillar 2—Agile Regulations

Although the use of animals for regulatory purposes is outside the scope of this article, for this pillar we offer some suggestions to explore how developing universally accepted standards and harmonising regulations could increase the use and acceptance of NAMs.

The need for common standards for NAMs is necessary for their implementation and for improving the confidence in them which will drive their wider adoption, and discussions around standardisation and qualification of NAMs are underway. In 2017, the International Consortium for Innovation and Quality in Pharmaceutical Development (IQ) microphysiological systems (MPS) Affiliate group was formed. One of the goals of this initiative is to qualify MPS—identifying contexts of use and defining the key characterisation data needed to allow the incorporation of MPS-derived data in pharmaceutical safety screening [57]. For biomedical ‘big data’, the lack of standardisation (in terms of ontology, terminology, data format, etc.) and the existence of (often incompatible) heterogeneous databases complicate the application of these data and prevent effective data sharing and data mining [58,59]. The Horizon 2020 programme STANDS4EU aims to “evaluate strategies for data integration and data-driven in silico modelling approaches to develop standards, recommendations and guidelines for personalized medicine” [60]. In silico approaches could include the use of databases, machine learning, artificial intelligence, molecular modelling, along with quantitative structure activity relationships and network analysis tools that permit the development, and crucially, subsequent testing of a model(s). There are additional efforts to apply in silico approaches in medical device and drug development: both the US Food and Drug Administration and the European Medicines Agency are developing guidelines for the use of in silico methods for regulatory purposes [61,62]. In 2021, the European Joint Research Centre, together with the European Standards Organisations CEN and CENELEC organised a workshop entitled “Putting Science into Standards—Organ on Chip: Towards Standardisation” as part of the Putting Science into Standards initiative, which aims to identify areas where standardisation could enable innovation of emerging technologies.

Global harmonisation of testing strategies, and standardisation of the tools, will be necessary to improve confidence in NAMs data and enable full replacement of animals. This will require some degree of flexibility in the regulations to adapt to, and accept data from, NAMs.

## 7. Pillar 3—Knowledge Transfer

The EU already supports several training and education initiatives. This includes structured collaborations between Member States such as the European Education Area or the European Research Area, the allocation of European Regional Development Funds to universities, or the creation of training courses launched by the Commission itself. These provide an effective vehicle to promote the acquisition of knowledge necessary for broader use, creation, development and application of NAMs.

We acknowledge the Commission’s ongoing efforts toward developing training and education resources for scientists as a first step towards the achievement of Pillar 3 by widening participation of key stakeholders through education, training and (re-)training [63]. However, these are centred around the 3Rs, and in order to facilitate a full, just transition, there needs to be a specific focus on replacement. This should include the inclusion of teaching on NAMs approaches across life sciences curricula at higher education, along with the introduction of the more human-relevant, innovative methods at secondary school level and possibly even earlier, as is being addressed for the 3Rs by the Commission with the European Schoolnet collaboration [64]. At the post-graduate level, more practical, hands-on training could be encouraged, perhaps even culminating in a formal qualification, equivalent to those currently used for animal handling [65,66]. In the UK, there is already some evidence of this shift, with the creation of Centres for Doctoral Training (CDT) dedicated to NAMs approaches for biomedical research. For example, lifETIME is the Engineered Tissues for Discovery, Industry and Medicine CDT, a partnership between the University of Glasgow, the University of Birmingham, Aston University and CÚRAM—Science Foundation Ireland. This aims to develop “bioengineered humanised 3D models, microfluidics, diagnostics and sensing platforms” in order to innovate biomedical research and drug discovery, offering formal training in these advanced, human-relevant tools and creating researchers with a clear understanding of the use and application of NAMs approaches (https://lifetime-cdt.org (accessed on 20 October 2021)).

We also note the efforts of projects coordinated by the Joint Research Centre that have resulted in the development of various valuable knowledge sources collating non-animal approaches for human diseases. These currently cover respiratory tract diseases [67], breast cancer [68], immuno-oncology [69], and neurodegenerative diseases [70]. These projects offer a snapshot of the state-of-the-art NAMs in use or under development and should be made available to researchers, project reviewers, ethical approval boards and even competent authorities responsible for approving animal research, as a valuable collection of methods that do not require animal use and therefore could contribute greatly to a reduction strategy without adversely impacting research (or researchers).

Of course, there is also a need for continuing education beyond graduation. The NAMs tools are evolving at a rapid pace, necessitating lifelong learning programmes of use not just to the researchers developing and using these tools, but also for grant reviewers, publishers, editors, educators and regulators. We see a facilitatory role for the learned societies in formalising and encouraging this within their continuing Professional Development programmes. There could be a requirement for a defined number of hours of “Innovation Engagement” to ensure familiarity with state-of-the-art NAMs methods as these continue to change, improve and are used across more fields of research.

Recently a highly cited paper has been analysed as having a worth of around 14,000 USD per annum [71]. Academic researchers exist under the cloud of “publish or perish”, and therefore we must ensure that they have confidence that a shift in methodology to the NAMs approaches will not preclude publication in these high-impact journals. However, there is an increasingly wider recognised issue—that “reviewer three” would like to see additional in vivo data as a condition of publication [72]. Any attempts to reduce animal use have to come with assurances that the careers of researchers will not suffer as a consequence of a shift. This requires coordinated efforts at the levels of grant reviewers, editors, peer reviewers, academic promotion boards, etc. Thus, there is a need for training and education in NAMs tools to provide expert input into grant, ethical and paper review, such that applications or papers that are entirely dedicated to NAMs approaches are not overlooked or “marked down” as a consequence of the inexperience of the reviewers regarding methods presented that prevents a valid critique of the science (Prof. L. Harries, personal communication, 12 December 2020). The expert body currently curated by the European Commission is ideally placed to offer this input [73].

Training and education—for every career stage and across all stakeholders—will be crucial to develop and maintain the workforce needed for the success of Pillar 3. There are many existing initiatives that could be implemented and expanded to achieve this.

## 8. Tracking Progress by Developing Metrics

In terms of basic and applied and translational research, and looking at data from 28 Member States only (without Norway), between 2015 and 2018, the average annual percentage decrease in animal use for basic, applied and translational research was 1.6%. Assuming that a 1.6% annual decrease could be sustained, total animal use would reach 50% of current levels by 2061 and would still be above one million uses per year by the year 2095. These are conservative estimates based on the Directive alone, and it is apparent that taking account of the rapid evolution in NAMs development would allow the Action Plan to accelerate the decline in animal use. We have adapted this to examine what three different (worst case, mid case and best case) entirely hypothetical scenarios would look like, using a linear decrease for simplicity, although we appreciate that this is unlikely to reflect the more complicated real-world picture (Figure 3). For the worst-case scenario, we transformed the average percentage annual decrease in use for basic, applied and translational research to the number of animals (100,000) and this gives a shallow decline in animal use such that use does not near zero till 2081. The mid-case scenario (grey line in Figure 3) is based on the average annual reduction in **total** animal use between 2015 and 2018 and represents a drop of 150,000 uses per year. Under these conditions, animal use reaches half the 2018 levels by 2039 and zero by 2060. The more ambitious best-case uses an annual reduction of 200,000 uses and here we see animal use halving by 2029 and getting to zero by 2040.

Using a linear projection is a simplistic view and it is perhaps more likely that a decline in animal use could be less straightforward and will reflect changes in circumstances, funding etc. For example, we have already seen that total animal use in Great Britain dropped by 15% in 2020, with over 30,000 fewer animal uses reported for basic, applied and translational research [74] as a consequence of national lockdowns due to the SARS-CoV-2 pandemic. It seems unlikely that this reduced animal use could be sustained “under normal circumstances” and indeed, it seems that many animals were culled as researchers could not access laboratories [75], rather than a sign that researchers are shifting away from animals toward non-animal approaches. Also, given that an annual reduction of 30,000 animal uses represents 32% of the total animal use for basic, applied and translational purposes in Great Britain alone. If this could be adopted across all member states then a 32% reduction equates to around two million uses annually and therefore would represent a far higher reduction target than even the best-case scenario presented in Figure 3 (200,000 animals a year, or around 3% total animal use) and may be too ambitious as a starting figure.

Additionally, compared to the current trend of an annual decrease of 1.6% of animal use, decreasing animal use by 150,000 or 200,000 animals per year represents a considerable “saving” of almost 26 million or 53 million animals, respectively, over the next three decades. Such long-term predictions are hypothetical, but given that over 60 years of “3Rs implementation” has yet to bring us anywhere near the ultimate goal of full replacement, they are worth exploring, in the context of an ambitious and pro-active EU action plan.

Some members of the pharmaceutical industry have already achieved substantial reductions in their use of animals for research and development and regulatory purposes. For example, Sanofi has estimated that since 2013, its animal use has decreased by about 45% [50]. Although this is one example, this is a much steeper decline than the one suggested above, and illustrates that dramatic reductions in animal use can be achieved in a short period of time with the appropriate strategy and commitment.

It is worth noting that reducing the number of animals used, in isolation, is not an indicator that NAMs have been more widely adopted. A drop in animal use could reflect, for example, extreme conditions whereby laboratories could not operate at full capacity due to a pandemic, a change in reporting requirements, or budget cuts, and this is particularly true where numbers decline for a one-year reporting period only. It is therefore important that we do not rely solely on animal use statistics, despite the phase out of animals being the ultimate goal; we must also consider other metrics, such as funding. These metrics could also exploit information collected as part of Pillars 1 and 3 to record education, training, continuing education, grant applications, publications etc. associated with a shift to NAMs methods. For example, it may be useful to report the number and monetary value of grants dedicated to NAMs, and even track the numbers of patents, publications, citations, etc. as measures of “success” for transitioning biomedical research to human-based NAMs. Defined metrics are therefore needed to monitor these developments and track their impact.

It is also true that, although this commentary is focused on the European Union, the activities and initiatives described in the Pillars could have global resonance. Animal use for research and testing is not solely a European issue. It is not possible to accurately quantify the numbers of animals used for scientific purposes globally, but with estimates varying from around 112 million in the United States alone [76], to 192.1 million globally in 2015 [77], it is clear that animals still bear the brunt of biomedical research and testing. Adoption of a phase-in approach across the European Union sends a clear signal that animal use is outdated and could help to drive global changes.

## 9. Concluding Remarks

The historic resolution from the European Parliament provides a call to action to revolutionise the health research paradigm in Europe, recognising human biology as the gold standard, and prioritising funding for the development and application of more predictive tools, based on human biology. We suggest three pillars of activity would be helpful to ensure that innovative science, education and training and regulatory flexibility are taken fully into account. We envision that a redirection of 10% of the EU’s annual research budget, increasing year on year, towards non-animal research employing NAMs, with a reduction of 200,000 animal uses per annum would bring animal use in basic, applied and translational research to a halt in around thirty years, whilst maintaining the EU’s thriving research environment. It is important to consider that phasing-out the use of animals in research is not only about taking animals out of the biomedical research paradigm, it requires the creation of a scientific environment where NAMs, such as microphysiological systems, computational modelling or ‘omics technologies, are accepted as the ‘new normal’ in laboratories; where researchers are equipped with the requisite skills to effectively apply these methods; and where research becomes human biology-focused. If this is accompanied by an increase in funding dedicated to NAMs as we describe, then the developments in science should keep pace with the reduction in animal use. Where poorly predictive animal models have been identified and effectively defunded, this frees-up funding to be shifted to human-relevant approaches. This offers a win-win for science and animals, adheres to the wishes of European citizens, and a better understanding of human biology should have an additional impact on drug discovery and development.

## Figures and Tables

**Figure 1 animals-12-00863-f001:**
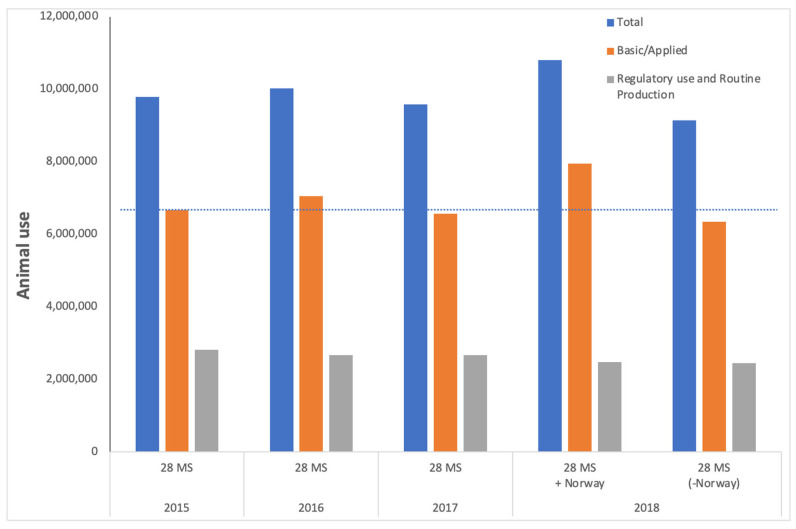
Animal use across the European Union is not undergoing any sustained decline. The blue bars indicate total animal use recorded for each year for all purposes, and according to the counting requirements defined in Directive 2012/63/EU and recorded in ALURES. The orange bars are combined animal use for Basic research and Translational and applied research and the grey bars represent the number of uses of animals for Regulatory use. Note that data from Norway are included in the ALURES database for the first time in 2018, creating an artificial increase in animal use. This is addressed with the data presented as the bars on the far right, which are from the 28 Member States (in direct comparison with the data from 2015, 2016, and 2017). The dotted line indicates the level of use for basic research and translational and applied research in 2015 and tracking this across to the bars on the far right illustrates the absence of any significant decrease in animal use for these purposes.

**Figure 2 animals-12-00863-f002:**
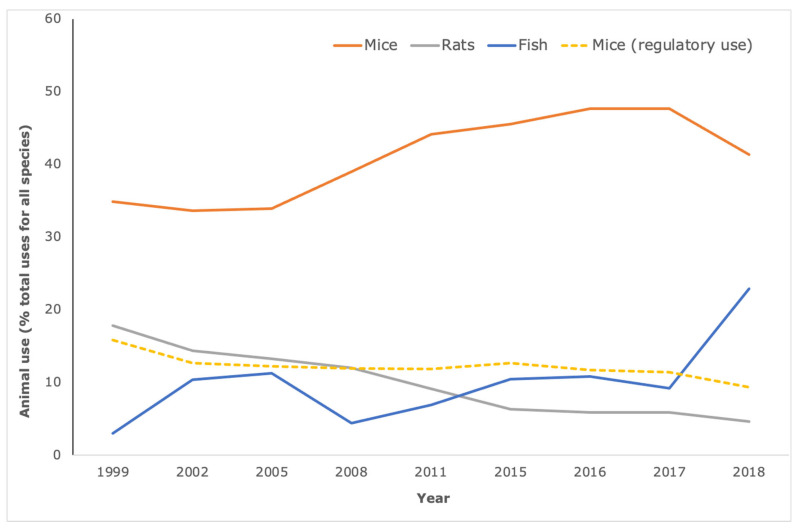
Mice comprise the majority species used in Basic Research and Translational & applied Research. According to the European Commission statistical reports on animal use, the species most commonly used for biomedical research are mice (orange line) rats (grey line) or fish (including zebrafish) blue line. For mice, this equates to over four million uses per year for these purposes, representing around 80% of the annual, total mouse use and around 50% of total animal use. In contrast, mouse use for regulatory use and routine production (yellow dotted line) is on the decline and is less than 10% of total animal use. Note that the increase in fish use for 2018 is attributable to the inclusion of data from Norway for the first time.

**Figure 3 animals-12-00863-f003:**
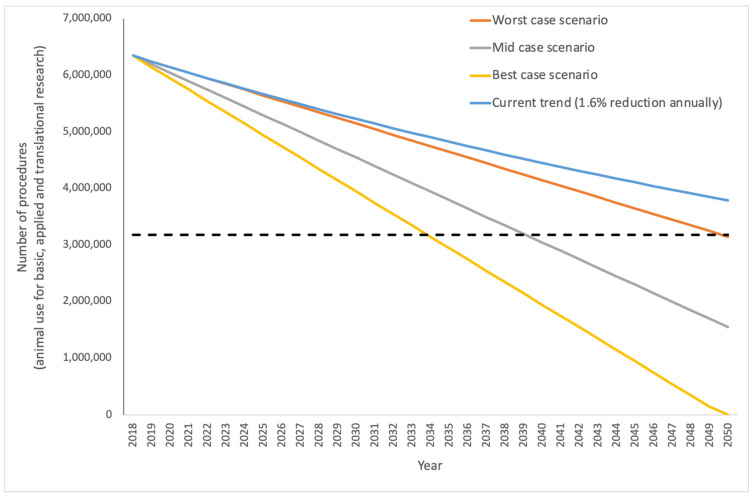
Projection of different reduction targets for animals used in biomedical research, based on a linear decrease. Beginning with the most recent data [7] where over 6.3 million animals were used for biomedical research across 28 Member States (without Norway), the blue line indicates the trajectory if animal use was reduced by 1.6% per year in accordance with current trends. For reference, the dashed line indicates the position of 50% of current, 2018 use (i.e., 317,899 animals). If an annual 1.6% decrease was maintained, the use of animals would not even be halved by 2050. We present three alternative scenarios: the orange line shows a ’worst case’- reduction of 100,000 animals per year; the grey line represents the ‘mid case’—reduction of 150,000 animals per year; and the yellow line shows the ‘best case’, adopting an annual reduction of 200,000 uses and here, animal use would reach zero by 2050.

**Table 1 animals-12-00863-t001:** Stakeholder identities, anticipated needs, and the pillar into which these activities may fit.

Pillar 1: Promoting Innovative Science with Human Biology as the Gold Standard	Pillar 2: Agile Regulations	Pillar 3: Knowledge Transfer	Stakeholder Group	Anticipated Needs
✓		✓	NAMs technology developers	Funding Training Infrastructure
	✓	✓	Regulators (drugs, chemicals and other regulated industry sectors)	Training Confidence in NAMs data (case studies) Expertise New tools to expedite approvals
✓		✓	Resource developers (NAMs databases, etc.)	Funding for curation/update Training in use and data deposition
✓		✓	Small and medium enterprises (NAMs-focused)	Funding Training Infrastructure
	✓	✓	Regulated community (i.e., animal users)	Training Confidence (case studies) Expertise
✓		✓	Grant reviewers	Training Expertise—targeted inclusion of NAMs experts for project review
✓		✓	Project reviewers	Training—more exposure to data from NAMs to enable confidence in accepting NAMs as major elements of research projects Case studies Expertise (pool of experts)
✓		✓	Ethical review boards	Training to increase confidence in NAMs Expertise—targeted inclusion of ethicists and NAMs users Resources Interaction with human ethical review boards
✓		✓	Researchers	Training Dedicated funding Infrastructure Confidence Secure career progression
✓		✓	Early career researchers	Confidence Secure career progression Incentives Centres of doctoral training focused on NAMs Commitment from funders—long term grant programmes
✓	✓	✓	Pharmaceutical companies	Training Funding incentives Confidence
	✓	✓	Contract research organisations	Training Incentives to enable business realignment
		✓	Life science students	Education Career path
		✓	Animal Care Technicians	Training/career realignment Infrastructure
		✓	(Human) Clinicians	Education—inclusion in parallel clinical trials Infrastructure
✓		✓	(Human) Patients	Engagement Education Collaboration
		✓	General public	Education Engagement
✓		✓	Learned societies	Engagement Collaboration Expertise
✓		✓	Instrumentation suppliers	Engagement Collaboration Incentives?
	✓	✓	Guidance documentation ICH, OECD, etc.	Collaboration Data sharing Expert input–guidance revision map to non-animal advances Harmonisation to prevent displacement of animal use

## Data Availability

Not applicable.

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
