# Peer review of "Phase-In to Phase-Out—Targeted, Inclusive Strategies Are Needed to Enable Full Replacement of Animal Use in the European Union"

_animals, 2022, doi:10.3390/ani12070863_

Round 1

Reviewer 1 Report

In this commentary, the authors discuss the current use of animals for research, testing and education within the EU, and suggest actions to promote the shift towards NAMs.

Even though I fully agree with the total phasing-out of animal models in biomedical research, I cannot help but feel rather underwhelmed by this manuscript and the lack of innovative ideas. Almost everything that the authors describe has been mentioned several times before: the overview of the number of animals and the temporal trends (Busquet et al. 2020; Dunn 2021; Langan & Brooks 2021; Taylor & Alvarez 2019), or the better relevance and prediction of human biology-based approaches (Golden et al. 2021; Marx et al. 2016; Shi et al. 2021). The recommendations are, once again, nothing new. Similar suggestions have been already proposed by, e.g., Hartung 2019; Hartung & Leist 2008; Herrmann et al. 2019; Prescott et al. 2017.

Moreover, there is a lot of repetition throughout the text.

Minor comments: 1. Throughout the manuscript, there are many random spaces between sentences. 2. The data in Table 1 would be better presented in a form of a figure/diagram.

References:

Busquet F, Kleensang A, Rovida C, Herrmann K, Leist M, Hartung T (2020) New European Union statistics on laboratory animal use – what really counts! ALTEX 37, 167-186.

Dunn R (2021) Brexit: A Boon or a Curse for Animals Used in Scientific Procedures? Animals 11.

Golden E, Macmillan DS, Dameron G, Kern P, Hartung T, Maertens A (2021) Evaluation of the global performance of eight in silico skin sensitization models using human data. ALTEX 38, 33-48.

Hartung T (2019) Research and Testing Without Animals: Where Are We Now and Where Are We Heading? In: Animal Experimentation: Working Towards a Paradigm Change, pp. 673-687. Brill.

Hartung T, Leist M (2008) Food for thought... on the evolution of toxicology and phasing out of animal testing. ALTEX 25, 91-102.

Herrmann K, Pistollato F, Stephens ML (2019) Beyond the 3Rs: Expanding the use of human-relevant replacement methods in biomedical research. ALTEX 36, 343-352.

Langan LM, Brooks BW (2021) Exploratory analysis of the application of animal reduction approaches in proteomics: How much is enough? ALTEX.

Marx U, Andersson TB, Bahinski A, Beilmann M, Beken S, Cassee FR, Cirit M, Daneshian M, Fitzpatrick S, Frey O (2016) Biology-inspired microphysiological system approaches to solve the prediction dilemma of substance testing. ALTEX 33, 272-231.

Prescott MJ, Langermans JA, Ragan I (2017) Applying the 3Rs to non-human primate research: barriers and solutions. Drug Discovery Today: Disease Models 23, 51-56.

Shi M, Wesseling S, Bouwmeester H, Rietjens I (2021) A new approach methodology (NAM) for the prediction of (nor) ibogaine-induced cardiotoxicity in humans. ALTEX 38, 636-652.

Taylor K, Alvarez LR (2019) An estimate of the number of animals used for scientific purposes worldwide in 2015. ATLA 47, 196-213.

Reviewer 2 Report

Overall, this is a clearly presented and worthwhile paper as there is a strong need  for a clear map with strategies to achieve replacement by a specific date in the not too distant future. 

As an ethicist who sits on Animal Ethics Committees in Australia, I support all your suggestions which also are much needed in Australia.

In addition I believe another Pillar (probably prior to your 3 pillars) is needed i.e. Prevention of Disease, with two areas of focus i.e. 1. Lifestyle changes, with Funding Support and Incentives to expedite behaviour changes to prevent disease, particularly in wealthy countries,  2. Environmental and Structural Support for poorer countries with Funding Support to overcome poverty and prevent the causes of disease.

The details you have provided for your 3 pillars and how they are being developed and can be developed further were well-organised and informative.

In Table 1, clarification on what you mean by "Regulated community" would be good - I'm not sure whether this means "General Public"? Certainly they are stakeholders.   Another possible category "Those with disease/illness" perhaps should be given education and information about animal use and alternatives to make informed choices  about the treatments they are offered so that they can encourage and support NAM development.

As a reader from Australia, it would be helpful to have brief explanations within the text of: Horizon Europe (Line 315), Framework Prog 7 and Horizon 2020 (Line 324), ring-fenced - an alternative word perhaps? (Line 336), and in silico approaches (Line 375/6). 

Only noticed 2 errors : Line 260 "use" to "used"; Line 264 add "of"

Good luck with your article and the promotion of these initiatives. 

Reviewer 3 Report

See attached

Reviewer 4 Report

This is an important and timely review, which addresses recent EU policy. I have two larger points and various minor suggestions (outlined below).

First, the manuscript’s recommendations could be clarified. This is achieved well in the Conclusion, but the take-home messages are a bit lost in the discussion of each pillar (L307-441). This could be improved by ending each section with a summary of clear and practical recommendations.

Second, for more general interest, the authors might consider a brief section near the end on the situation in the rest of the world. The EU focus is valuable, but this is not an EU-only issue. I think the authors’ recommendations have potential global applications.

Simple Summary:

  • L9-12: Grammar? This sentence feels like it’s building to a verb/object, which never comes.

Introduction

  • L36-42: Each of these points could be explored in more detail, and backed up with stronger evidence. How many animals were used six decades ago versus now? What are the details of Article 4? How have opinion polls shifted in various EU countries? How has the “development and application of” NAMs been “almost exponential”?
  • L44-45: Is it “scientifically possible to do so” now? If not, there isn’t necessarily a contradiction.

The current situation - sustained reliance on animals across the research spectra

  • L66-69: I would encourage the authors to justify this statement with reference to the “available data”.
  • L76: Grammar.
  • L84: It would be interesting to break this down into different species/taxa (this is only done incompletely from L124 and in Figure 2).
  • L103-106: In my view, without accounting for total research output, this is (potentially) an overly simplistic interpretation. If total research output has increased, maintaining roughly the same level of animal experimentation would represent a “real terms” reduction. Are data on total EU research output available?
  • L126-127: Statements like this need clarifying and backing up with evidence. Clearly, mice are more “human relevant” than, say, C. elegans.
  • L124: Sorry to be picky, but neither rodents nor fish are “species”.
  • L125: I’d recommend changing the wording to “…of SCIENTIFIC tractability…” since genetics aren’t the only thing it’s very easy to study in mice (e.g., development and the brain are also very well characterised).

Transitioning biomedical research towards human-based, non-animal methods represents an essential step in achieving the EU’s public health objectives

  • L230: Ref?
  • L237: Is this “likely”? What’s the evidence?

Planning an inclusive transition to non-animal research

  • L271-274: This doesn’t really seem relevant. It’s also not an “evolutionary advance” in the sense that it’s not (presumably) underpinned by changing gene frequencies in the human population.
  • L300: It’s at least up for debate, I think, whether animals are the stakeholder with the most to lose. In particular, I’m thinking about human patients with chronic or life-threatening conditions.
  • L305: As above, this table should probably include human patients that research is intended to benefit. Also, unless I’m missing something, please clarify what “ü” and “P” mean.

Pillar 1 - Promoting innovative science with human biology as the gold standard

  • L333: How would “poorly predictive animal models” be defined?

Pillar 2: - Agile regulations

  • L359: “outwith” > “outside”.
  • L367: Please define this acronym (Microphysiological Systems).

Tracking progress by developing metrics

  • L463: What factors underpinned this decline? How was it achieved? What happened to total research output? If explained in more detail, this example could be an informative case study.

Reviewer 5 Report

The paper is perfect, and it can be ready for publication.

I only have a couple of comments, that are not necessary to the relevance of the paper that can be published as is.

  1. Figure 1 lists the number of animals used across the EU. The caption specifies that the bars are about the total animal use recorded "as defined in Directive 2012/63/EU". This is not fully true because the Directive asks to consider all animals, including puppies and satellite animals while these animals are not counted in the statistics (https://doi.org/10.14573/altex.2003241)
  2. The statistics consider that many tests are performed outside the EU, but no details are provided in the reports of the Commission
  3. Table 1, Make up: I would move the column “stakeholder group” in first position. under the three pillars there are strange flags, but probably it is a mistake of the format in the downloaded paper. 
  4. It is strange that ICH guidelines are mentioned nowhere in the paper (just a row in table 1), in particular in the paragraph about Pillar 2 -Agile regulation
  5. In a couple of sentences, it is mentioned that the EU is composed by 28 member states. Unfortunately, they are now only 27. UK is still counted in the statistics of animal used for scientific purposes. "(without Norway)" should be replaced by "(without Norway and including UK)"
  6. The total number of animals used in research will never get to zero if you’re considering also the application in veterinary medicine.

Round 2

Reviewer 1 Report

The authors failed to address my comments from the previous round of review, namely the lack of novelty and repetitive passages. Even the minor point - double (?) spacing between most of the sentences - has not been resolved.

Furthermore, having read the text for the second time, what stood out to me was the lack of discussion on scientific data comparing outcomes of animal experiments and NAMs. Lastly, as another minor point, I am wondering why the authors assume in Figure 3 that the reduction would have a linear trend.

Reviewer 3 Report

See attached

Reviewer 4 Report

Thank you for addressing my comments - I really enjoyed the revised manuscript!

Author Response

We thank the reviewer very much for their support in improving the manuscript and for their kind words.